# Black-Box Ripper: Copying black-box models using generative evolutionary algorithms

**Antonio Bărbălău**[1,*]**, Adrian Cosma**[2]**, Radu Tudor Ionescu**[1]**, Marius Popescu**[1]
[1]University of Bucharest, Romania
[2]University Politehnica of Bucharest, Romania
*abarbalau@fmi.unibuc.ro

## Abstract

We study the task of replicating the functionality of black-box neural models, for which we only know the output class probabilities provided for a set of input images. We assume back-propagation through the black-box model is not possible and its training images are not available, e.g. the model could be exposed only through an API. In this context, we present a teacher-student framework that can distill the black-box (teacher) model into a student model with minimal accuracy loss. To generate useful data samples for training the student, our framework $(i)$ learns to generate images on a proxy data set (with images and classes different from those used to train the black-box) and $(ii)$ applies an evolutionary strategy to make sure that each generated data sample exhibits a high response for a specific class when given as input to the black box. Our framework is compared with several baseline and state-of-the-art methods on three benchmark data sets. The empirical evidence indicates that our model is superior to the considered baselines. Although our method does not back-propagate through the black-box network, it generally surpasses state-of-the-art methods that regard the teacher as a glass-box model. Our code is available at: `https://github.com/antoniobarbalau/black-box-ripper`.

## 1 Introduction

In the last couple of years, AI has gained a lot of attention in industry, due to the latest research developments in the field, e.g. deep learning [21]. Indeed, an increasing amount of companies have started to integrate neural models into their products [23], which are used by consumers or third-party developers [11]. In order to protect their intellectual property, companies try to keep information about the internals (architecture, training data, hyperparameters and so on) of their neural models confidential, exposing the models only as black boxes: data samples in, predictions out. However, recent research [17, 30, 31, 32, 35, 36] showed that various aspects of black-box neural models can be stolen with some effort, including even their functionality.

Studying ways of stealing or copying the functionality of black-box models is of great interest to AI companies, giving them the opportunity to better protect their models through various mechanisms [12, 40]. Motivated by this direction of study, we propose a novel generative evolutionary framework able to effectively steal the functionality of black-box models. The proposed framework is somewhat related to knowledge distillation with teacher-student networks [2, 10, 22, 39], the main difference being that access to the training data of the teacher is not permitted to preserve the black-box nature of the teacher. In this context, we train the student on a proxy data set with images and classes different from those used to train the black-box, in a setting known as zero-shot or data-free knowledge distillation [1, 3, 4, 8, 24, 28, 38]. To our knowledge, we are among the few [17, 31] to jointly consider no access to the training data and to the model's architecture and hyperparameters, i.e. the model in question is a complete black-box.

As shown in Figure 1, our framework is comprised of a black-box teacher network, a student network, a generator and an evolutionary strategy. The teacher is trained independently of the framework,

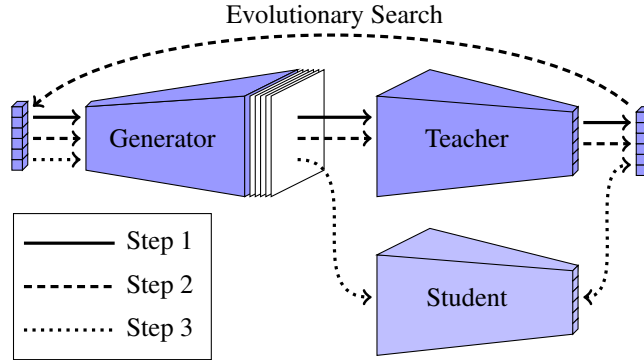

Figure 1: Step 1: Generate a set of random samples in the latent space and forward pass them through the generator and the teacher. Step 2: Optimize the latent space encodings via evolutionary search. Step 3: Distill the knowledge using the optimized data samples as input and the teacher's class probabilities as target. Gradients are propagated only through the student network.

being considered a pre-trained black-box model from our point of view. In order to effectively steal the functionality of the black box, our framework is based on two training phases. In the first phase, we train a generative model, e.g. a Variational Auto-Encoder (VAE) [16] or a Generative Adversarial Network [7], on the proxy data set. In the second training phase, we apply an evolutionary strategy, modifying the generated data samples such that they exhibit a high response for a certain class when given as input to the teacher.

To demonstrate the effectiveness of our generative evolutionary framework, we conduct experiments on three benchmark data sets: CIFAR-10 [18] with CIFAR-100 [18] as proxy, Fashion-MNIST [37] with CIFAR-10 [18] as proxy and 10 Monkey Species [27] with CelebA-HQ [13] and ImageNet Cats and Dogs [33] as proxies. We compare our framework with a series of state-of-the-art methods [1, 28, 31], demonstrating generally superior accuracy rates, while preserving the black-box nature of the teacher. For example, on CIFAR-10 as true data set and CIFAR-100 with 6 classes as proxy data set, we surpass the state-of-the-art performance by a significant margin of $10.4\%$. We also include ablation results, showing the benefits of adding our novel component: the evolutionary algorithm.

In summary, our contribution is twofold:

- We propose a novel generative evolutionary algorithm for stealing the functionality of black-box classification models.
- We demonstrate that our framework is significantly better than a state-of-the-art method [31] trained in similarly adverse conditions (Orekondy et al. [31] consider the teacher as a black box) and equally or slightly better than a state-of-the-art method [1] trained in relaxed conditions (Addepalli et al. [1] back-propagate gradients through the teacher network).

## 2 Related Work

Our work is related to zero-shot knowledge distillation methods [1, 3, 4, 8, 24, 28, 38], with the difference that we regard the teacher model as a black box, and to model stealing methods [17, 30, 31, 32, 35, 36], with the difference that we focus on accuracy and not on minimizing the number of API calls to the black box.

**Zero-shot knowledge distillation.** After researchers introduced methods of distilling information [2, 10] from large neural networks (teachers) to smaller and faster models (students) with minimal accuracy loss, a diverse set of methods have been developed to improve the preliminary approaches, addressing some of their practical limitations for specific tasks. A limitation of interest to us is the requirement to access the original training data (of the teacher model). Many formulations have been developed to alleviate this requirement [1, 8, 4, 24, 28], with methods either requiring a small subset of the original data [3, 4], or none at all [1]. Nayak et al. [28] proposed a method for knowledge distillation without real training data, using the teacher model to synthesize data impressions via back-propagation instead. While the generated samples do not resemble natural images, the student is able to learn from the high response patterns of the teacher, showing reasonable generalization accuracy. Methods to synthesize samples through back-propagation, e.g. feature visualization methods, have gained a lot of interest in the area of knowledge distillation. Nguyen et al. [29] showed that, through network inversion, resulting feature visualizations exhibit a high degree of realism. Further, Yin et

al. [38] used the same method to generate samples for training a student network, while employing a discrepancy loss in the form of Jensen-Shannon entropy between the teacher and the student. While showing good results, these methods are not considering the teacher model as a black box, since back-propagation implies knowledge of and access to the model's weights. Micaelli et al. [24] developed a method for zero-shot knowledge transfer by jointly training a generative model and the student, such that the generated samples are easily classified by the teacher, but hard for the student. In a similar manner to Yin et al. [38], a discrepancy loss is applied in the training process between the teacher and the student. We take a different approach, as our generative model is trained beforehand, and we optimize the synthesized samples through evolutionary search to elicit a high response from the teacher. More closely-related to our work, Addepalli et al. [1] proposed a data-enriching GAN (DeGAN), that is trained jointly with the student, but on a proxy data set, different from the model's inaccessible, true data set. The generative model generates samples such that the teacher model outputs a confident response, through a loss function promoting diversity of samples and low-entropy confidence scores. In the context of their framework, by means of back-propagation though the teacher network, the GAN is able to synthesize samples that help the student approach the teacher's accuracy level. Different from their approach, we do not propagate information through the teacher, as we consider it a black-box model. Moreover, our generative model is fixed, being trained a priori on a proxy data set, which is not related to the true set. Unlike Addepalli et al. [1] and all previous works on zero-shot knowledge distillation, we generate artificial samples through evolutionary search, using these samples to train the student.

**Model stealing.** While stealing information from deep learning models has been studied in a setting where the architecture and parameters are available to an attacker, the more realistic setting in which the model is available through an API, being considered a black box (data samples in, probabilities out), still remains a difficult problem. Oh et al. [30] proposed a framework to infer meta-information of a neural network API, in the form of architectural design choices, optimization algorithm and the type of training data. Wang et al. [36] showed that the hyperparameters of popular machine learning models deployed in predictive analytics platforms can be inferred by solving a set of hyperparameter equations, derived through either the model parameters, or through querying the model given the training data set. Unlike these methods, we aim at stealing the functionality of the black-box model, reproducing its results as best as possible. Papernot et al. [32] developed a method of estimating the decision boundary of a black-box API with the goal of crafting adversarial attacks. While their technique of using Jacobian-based data set augmentation is not aimed at offering high accuracy to their student model, it is clear that synthetic generation of input samples in the absence of original training data is a popular method of training substitute models. Instead of manipulating Gaussian noise, we address the problem of synthetic data generation by employing a proxy data set, which is agnostic to the real training data used by the model, giving visual and semantic structure to the synthesized samples. Several methods for model extraction from APIs have been developed in the past [17, 31, 35], with good results on conventional machine learning models [35], natural language services [17] based on BERT [5] and computer vision models [31]. Closest to our setting, Orekondy et al. [31] developed an efficient method of querying a image classification API, to maximize the accuracy of the model copy while having a minimal amount of consumed resources. However, our setting is less strict, making fewer assumptions of the operating environment of the black box. Our data-free approach implies no knowledge of the black-box training data, while showing very good performance when using a visually and semantically unrelated proxy data set for the generative model. As such, we are interested in maximizing the accuracy of our student model, regardless of the number of queries.

## 3 Method

**Problem statement.** We approach the task of stealing the functionality of a classification model, while assuming no access to the model's weights and hyperparameters and no information about the training data. With these assumptions, the classification model is a black box. Our problem formulation is related to zero-shot knowledge distillation [1] and model functionality stealing [31]. In zero-shot knowledge distillation, a student model is trained to mimic a teacher model, while having full access to the teacher model, but no information about the training data (the student is trained on a so-called *proxy* data set). We consider a restricted setting of zero-shot knowledge distillation, in which there is no access to the internals (weights, hyperparameters, architecture) of the black box. In model functionality stealing, the black-box model is only accessible through a paid API, the goal

being that of maximizing student accuracy while reducing the number of model calls. We consider a relaxed setting of model functionality stealing, in which the number of model calls is not an issue, our primary focus being the accuracy level.

Let $T$ be a black-box classification model with parameters $\theta_T$ and $S$ a student classification model with parameters $\theta_S$. Our problem statement can be formally expressed as follows:

$$\min_{\theta_S} \|S(Z\downarrow\uparrow,\ X'\downarrow) - T(X\downarrow\uparrow,\ X'\downarrow)\|, \tag{1}$$

where $X$ is the original (or true) training set, $X'$ is the original (or true) test set and $Z$ is the proxy training set. In our study, we consider the realistic scenario in which $X$, $X'$ and $Z$ are disjoint sets. The notation $M(U\downarrow\uparrow,\ W\downarrow)$ symbolizes that the model $M$ is trained on a set $U$ and applied on a set $W$, i.e. the arrows indicate forward and backward passes through the model on the respective data set. The goal is to learn the parameters $\theta_S$ on the proxy data set $Z$ in order to reproduce the output probabilities of the teacher on $X'$, as best as possible. We note that the true test set $X'$ is not available during the optimization of $\theta_S$, being used only for evaluation purposes.

**Black-Box Ripper.** Noting that the proxy data set $Z$ has no data samples or classes in common to the true data set $X$, the task of training the student model directly on the data set $Z$ is very difficult, especially if $Z$ is small. We therefore propose a generative evolutionary framework, called Black-Box Ripper, to train the student model towards the optimization problem defined in Equation (1). Our framework enables us $(i)$ to generate as many examples as required for the complete convergence of the student and $(ii)$ to evolve the generated samples such that the discrepancy between the proxy data set and the true data set is minimized.

Our framework relies on a generative model to synthesize realistic samples for the student model. Although in practice we observed better results when the generator is a GAN [7], in theory, we could use any generative model, including a GAN or a VAE [16]. Since we assume no knowledge of the training data of the teacher model, we train our generator on the proxy data set, just as the student model. Unlike Addepalli et al. [1], we do not train the generator jointly with the student and we do not back-propagate gradients through the teacher. In our framework, we consider that the generator is trained in a preliminary step, independently of the student. This allows us to plug-in any generative model, including pre-trained models.

As shown in Figure 1, our framework is comprised of three steps. The first step is to generate a set of random samples in the latent space of the generator, passing them through the generator and obtaining a set of generated data samples $Z'$, such that $p_{Z'} \approx p_Z$, where $p_{Z'}$ and $p_Z$ are probability densities of the generated data $Z'$ and of the real (proxy) data $Z$, respectively. Model calls are performed afterwards in order to obtain a set of class probabilities from the teacher $T$. These class probabilities are to be used in the second step based on evolutionary optimization.

Since the generator is trained to model the probability density $p_Z$ and $p_Z \neq p_X$, the data samples are likely not representative for any class in the true data set $X$. Hence, the teacher is likely not going to produce a high probability for a certain class. The same problem occurs when there is no generator and the teacher is applied directly on images from $Z$. To this end, we introduce the second step in our framework, which is based on an evolutionary strategy to overcome the discrepancy between the proxy data set and the true data set, searching for samples in the latent space of the generator that elicit a high response from the teacher model towards a certain class.

Our use of the evolutionary algorithm is motivated by the assumption that the low-dimensional data manifold modeled by the generator is continuous, even though many data sets lie in disconnected low-dimensional manifolds of GANs [14], for example. While this is an undesirable property of GANs, resulting in artifacts at class boundaries, we leverage the continuity assumption to optimally traverse the latent space. Therefore, when training the student, we randomly select a class label and traverse the latent space of the generator $G$, minimizing the difference between the selected class label $y$ and the teacher's output $\hat{y}$ on the generated image. The fitness of a latent space vector $v$ can be evaluated only after passing it through the generator and the black-box teacher to obtain the corresponding class probabilities. The optimization process aims at generating images that are classified in class $y$ with high confidence by the teacher $T$. Formally, the objective $V$ of our evolutionary algorithm is the mean squared error between $y$ and $\hat{y}$. Hence, we aim to solve:

$$\min_{v} V(v, y, T, G) = \min_{v} \sum_{i=1}^{n} \left(T(X\downarrow\uparrow,\ G(Z\downarrow\uparrow, v\downarrow)\downarrow) - y_i\right)^2 = \min_{v} \sum_{i=1}^{n} (\hat{y}_i - y_i)^2, \tag{2}$$

where $v$ is a latent space vector and $n$ is the number of classes in $X$, the other notations being explained above.

---

**Algorithm 1** Evolutionary Optimization Algorithm

---

**Input:** $y$ - desired class label, $T$ - black-box teacher, $G$ - generator trained on $Z$.

**Hyperparameters:** $K$ - population size, $k$ - elite size, $u$ - latent space boundary, $t$ - threshold for stopping criterion.

**Output:** $p^*$ - generated data sample with high confidence on desired class $y$.

1: **procedure** OPTIMIZE($y, T, G, K, k$)
2:      Initialize population from a uniform distribution: $P \leftarrow \{U(-u, u)\}_K$
3:      Select fittest latent vector: $p^* \leftarrow min_{p \in P} V(p, y, T, G)$
4:      **while** $V(p^*, y, T, G) \geq t$ **do**
5:          Select fittest $k$ vectors: $P_e \subset P$
6:          Uniformly sample $K - k$ copies from $P_e$: $P_c \leftarrow \{\mathcal{U}(P_e)\}_{K-k}$
7:          Mutate copied vectors with Gaussian noise: $P_c \leftarrow P_c + \mathcal{N}(0, 1)$
8:          Replace old population with new one: $P \leftarrow P_e \cup P_c$
9:          Select fittest latent vector: $p^* \leftarrow min_{p \in P} V(p, y, T, G)$
10:     **return** $G(Z{\downarrow}{\uparrow}, \ p^*{\downarrow})$

---

Our evolutionary training strategy is formally presented in Algorithm 1. Latent space traversal starts by generating an initial population of $K$ latent space vectors (step 2). Until the value of the objective $V$ for the fittest latent vector $p^*$ is lower than a threshold $t$, we select the $k$ fittest individuals (step 5), replicate them (step 6), mutating the replicated vectors using random Gaussian noise (step 7). The algorithm outputs the data sample corresponding to the fittest latent vector $p^*$ (step 10).

Once we obtain mini-batches of data samples optimized through Algorithm 1, we proceed with our third and last step. We distill the knowledge of the teacher into the student model by minimizing the cross-entropy between the class probabilities of the teacher and the class probabilities of the student:

$$\mathcal{L}(\theta_S, Z') = - \sum_{z' \in Z'} T(X{\downarrow}{\uparrow}, \ Z'{\downarrow}) \cdot log(S(Z'{\downarrow}{\uparrow}, \ Z'{\downarrow})), \tag{3}$$

where $Z'$ is a set of images generated by our generative evolutionary algorithm from the proxy data set $Z$ and $z'$ is a data sample in $Z'$, the other notations being explained above.

## 4 Experiments

### 4.1 Datasets

We first evaluate our framework on a similar setting to that of Addepalli et al. [1], comparing our framework to relevant baselines and ablated versions of our own framework. These experiments include a couple of data set pairs, namely: CIFAR-10 as true data set with CIFAR-100 [18] as proxy data set, and Fashion-MNIST [37] as true data set with CIFAR-10 [18] as proxy data set. The latter pair of data sets features high visual discrepancy, given that Fashion-MNIST contains only grayscale images of 10 classes of fashion items, while CIFAR-10 contains natural objects in context.

Since CIFAR-10, CIFAR-100 and Fashion-MNIST have low resolution images, we also test our approach in a more realistic scenario with high-resolution images. For this set of experiments, we use the 10 Monkey Species [27] data set, containing images of 10 species of monkeys in their natural habitat, as true data set. In this scenario, we independently consider two proxy data sets, namely CelebA-HQ [13] and ImageNet Cats and Dogs [33]. CelebA-HQ contains high-resolution images of $1024 \times 1024$ pixels. ImageNet Cats and Dogs is composed of 143 species of cats and dogs. For the latter proxy, we additionally provide qualitative results to showcase our optimization process.

### 4.2 Baselines

**Training on proxy data (Knockoff Nets [31]).** Current methods focusing on stealing the functionality of black-box models rely on training the student on samples taken directly from the proxy data set, using labels provided by the teacher as ground-truth for the student. Since the most recent work in this category is based on Knockoff Nets [31], we include this relevant baseline in the experiments. For a fair comparison, we allow Knockoff Nets to do as many forward passes as necessary through the teacher, until the student reaches complete convergence.

Table 1: Accuracy rates (in %) on CIFAR-10 of various zero-shot knowledge distillation [1, 28] and model stealing [31] methods versus Black-Box Ripper. For our model, we report the average accuracy as well as the standard deviation computed over 5 runs. Best results are highlighted in bold.

| Proxy Dataset | CIFAR-100 90 classes | CIFAR-100 40 classes | CIFAR-100 10 classes | CIFAR-100 6 classes |
|---|---|---|---|---|
| Teacher Accuracy | 82.5 | 82.5 | 82.5 | 82.5 |
| Knockoff Nets [31] | 74.5 | 65.7 | 46.6 | 36.4 |
| ZSKD [28] | 69.5 | 69.5 | 69.5 | 69.5 |
| DeGAN [1] | **80.5** | 76.3 | $72.6 \pm 3.3$ | 59.5 |
| Black-Box Ripper (**Ours**) | $79.0 \pm 0.2$ | $\mathbf{76.5} \pm 0.1$ | $\mathbf{77.9} \pm 0.3$ | $\mathbf{69.9} \pm 0.2$ |

**Zero-Shot Knowledge Distillation (ZSKD [28]).** We include the results of Nayak et al. [28] as one of the baselines, as their data-free approach is very popular in this area of research. However, the authors directly synthesize data samples by back-propagating gradients through the teacher, using data visualization methods. Since they consider a more relaxed setting in which the teacher is a white box, their approach does not need a proxy data set.

**Data-enriching GAN (DeGAN [1]).** Even tough our work focuses on knowledge distillation in a realistic scenario, in which training data, structure and parameters of the teacher are completely obscured, we aim to compare our method to top white-box approaches in order to perform a more solid evaluation for Black-Box Ripper. We therefore consider DeGAN [1] as baseline. DeGAN employs a generative network trained in tandem with the student, while performing back-propagation through the teacher network, thus requiring complete access to the teacher. DeGAN is a very recent method, attaining state-of-the-art results in data-free knowledge distillation.

**Training on generator samples (GAN, VAE).** In order to show the improvement brought by the evolutionary optimization algorithm, we perform ablation experiments using the same generator as in Black-Box Ripper, but without applying the evolutionary strategy. Samples are generated by uniformly sampling from the latent space of the generator. As generative model, we considered two variants: a GAN and a VAE. Observing that GANs seem more useful for the student, we report results with VAE in a single experiment, applying the same rule to Black-Box Ripper.

## 4.3   Results on CIFAR-10

**Experimental setup.** We conduct experiments following Addepalli et al. [1], thus employing an AlexNet [19] architecture for the teacher and a half-AlexNet architecture for the student. All models, including baselines, are trained for 200 epochs using the Adam [15] optimizer. We used mini-batches of 64 images. In Black-Box Ripper, the images are synthesized by the evolutionary algorithm, using at most 10 iterations, a population of $K = 30$ latent vectors sampled within the boundary $u = 3$, an elite size of $k = 10$, halting the optimization if the fittest latent vector gives an objective value lower than $t = 0.02$. All other experiments on Fashion-MNIST and 10 Monkey Species use the same hyperparameters for our evolutionary strategy. Since the 10 classes in CIFAR-10 need to be removed from CIFAR-100, the maximum number of classes from CIFAR-100 that can be used is 90. As Addepalli et al. [1], we present results on four different proxies with 6, 10, 40 and 90 classes from CIFAR-100, respectively. As generators, we employ a Progressively Growing GAN (ProGAN) [13] for the 6 and 10 classes setup and a Spectral Normalization GAN (SNGAN) [26] for the other two experiments. Readers are referred to Addepalli et al. [1] for a detailed description of the experimental setup on CIFAR-10.

**Results.** We present results on CIFAR-10 as true data set in Table 1. For reference, the teacher's accuracy is 82.5%, this being the upper bound for Black-Box Ripper and other methods [1, 28, 31]. Even though we consider the teacher as a black box, our model manages to outperform the white-box DeGAN [1] in three out of four cases, while achieving a close result in the unfavorable case. These results indicate that our method does not need full access to the teacher in order to achieve state-of-the-art results in zero-shot knowledge distillation. The results also show that our optimization procedure brings more and more value (the improvements with respect to the baselines are higher) as the number of classes in the proxy data set gets smaller. We note that the CIFAR-100 proxies with less classes also contain classes that are more distant from the CIFAR-10 classes. When the generator does not benefit from a large variation of the proxy data set and the discrepancy between the proxy

Table 2: Accuracy rates (in %) of various zero-shot knowledge distillation [1, 28] and model stealing [31] methods versus Black-Box Ripper on Fashion-MNIST as true data set and CIFAR-10 as proxy data set. Ablation results with two generators, a VAE and a SNGAN [26], are also included. Best results are highlighted in bold.

| Architectures | VGG-16 | LeNet → Half LeNet |
|---|---|---|
| Teacher Accuracy | 94.2 | 89.9 |
| Knockoff Nets [31] | 82.9 | 77.8 |
| ZSKD [28] | - | 79.6 |
| DeGAN [1] | - | **83.7** |
| VAE (no evolutionary optimization) | 78.3 | 73.1 |
| SNGAN (no evolutionary optimization) | 87.6 | 80.0 |
| Black-Box Ripper with VAE **(Ours)** | 86.1 | 78.8 |
| Black-Box Ripper with SNGAN **(Ours)** | **90.0** | 82.2 |

and the true data set is larger, our evolutionary algorithm can help to close the distribution gap. This explains why Black-Box Ripper is $10.4\%$ over DeGAN on CIFAR-100 with 6 classes and $5.3\%$ over DeGAN on CIFAR-100 with 10 classes. Considering the small standard deviations computed over 5 runs, we conclude that our evolutionary strategy provides very stable results. We therefore report results for a single run in the subsequent experiments, just as Addepalli et al. [1].

## 4.4 Results on Fashion-MNIST

**Experimental setup.** Using Fashion-MNIST as a true data set and CIFAR-10 as proxy data set, we evaluate our framework in comparison to a set of relevant baselines, some being also considered by Addepalli et al. [1]. On Fashion-MNIST, we report results with two alternative generators in Black-Box Ripper, a VAE with the same architectural design as in [13] and a SNGAN [26]. We also present ablation results with these generators, eliminating the evolutionary strategy from the pipeline. For the teacher and the student, we consider two architectural choices: VGG-16 [34] for both networks, and LeNet [20] and half-LeNet for the teacher and student, respectively.

**Results.** We show results on Fashion-MNIST as true data set in Table 2. First, we note that using a deeper architecture (VGG-16 versus half-LeNet) plays an important role in reducing the accuracy gap with respect to the corresponding teacher. We surpass all baselines for the VGG-16 architecture, and report the second best result (just $1.5\%$ below DeGAN [1]) for LeNet. The ablations results with VAE versus Black-Box Ripper with VAE indicate that the evolutionary algorithm brings improvements of over $5\%$. Meanwhile, the differences between SNGAN and Black-Box Ripper with SNGAN are just over $2\%$. In both cases, there is a clear advantage in employing our evolutionary strategy.

## 4.5 Results on 10 Monkey Species

**Experimental setup.** As generators, we employed a pre-trained ProGAN [13] on CelebA-HQ and a pre-trained cGAN with projection discriminator [25] on ImageNet Cats and Dogs. As in the previous experiment, we include ablation results with GANs, excluding the evolutionary search from our pipeline. The teacher and the students are ResNet-18 [9] models. The teacher is trained for 30 epochs and the students are trained for 200 epochs using the same mini-batch size as the teacher.

**Results.** We present the results on 10 Monkey Species in Table 3. Our method yields a great performance improvement over Knockoff Nets [31] on CelebA-HQ as proxy, amounting to $14.7\%$. With respect to the student trained on images generated by ProGAN, the student trained with Black-Box Ripper (with ProGAN as generator) has a significant improvement of $12.1\%$.

**Qualitative Results.** We illustrate qualitative results of our evolutionary optimization process in Figure 2. In the top row of Figure 2, we show a selected evolutionary search process for the *common squirrel monkey* class in the 10 Monkey Species data set. The majority of *common squirrel monkey* instances in the data set often depict the monkey on tree branches. Meanwhile, the GAN is trained on images of cats and dogs, which appear in a different contextual environment. We therefore observe that the optimization process converges to an image of a cat with leaves and tree branches around it. The teacher gives a high response for this generated image (the confidence score for *common squirrel monkey* is $99.37\%$). In the bottom row of Figure 2, we show an example of evolutionary search for

Table 3: Accuracy rates (in %) of a state-of-the-art model stealing method [31] versus Black-Box Ripper on 10 Monkey Species as true data set and CelebA-HQ and ImageNet Cats and Dogs as proxy data sets. Ablation results with GANs are also included. Best results are highlighted in bold.

| Proxy Dataset | CelebA-HQ | ImageNet Cats and Dogs |
|---|---|---|
| Teacher Accuracy | 77.4 | 77.4 |
| Knockoff Nets [31] | 54.7 | - |
| GAN (no evolutionary optimization) | 57.3 | 65.0 |
| Black-Box Ripper **(Ours)** | **69.4** | **71.6** |

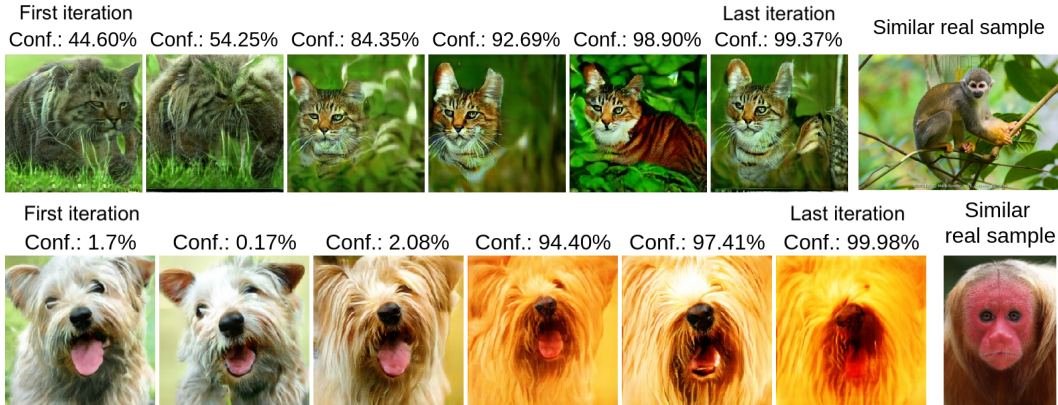

Figure 2: Top row: Evolutionary optimization progression from a *cat* from the proxy data set into a *squirrel monkey* class from the true data set. Bottom row: Progression from a *dog* from the proxy data set into a *bald uakari* from the true data set. Only the best specimen from the population at each iteration is shown.

the *bald uakari* monkey species. This species can be generally described as presenting orange fur and bald head with a red face. We note that our final (converged) image does not really resemble a monkey, but the teacher gives a very high response for the target class (the confidence score is 99.98%). The evolutionary search produces an image with close texture resemblance and coarse semantic resemblance. For example, the dog's snout is transformed into a red stain which is supposed to be the monkey's face. This observation is consistent with the properties of CNNs uncovered by Geirhos et al. [6], indicating that CNNs tend to recognize objects mostly based on texture patterns. We note that the images generated from the proxy data set distribution and aligned using evolutionary search contain sufficient high-level information for the student to copy the functionality of the teacher, although the generated images are visually different than the real images found in the true data set.

## 5  Conclusion

We proposed a novel black-box functionality stealing framework able to achieve state-of-the-art results in zero-shot knowledge distillation scenarios. We have tested our method on Fashion-MNIST, CIFAR-10 and 10 Monkey Species, using multiple proxy data sets and settings. We compared our framework with state-of-the-art data-free knowledge distillation [1, 28] and model stealing [31] methods. Our approach was able to surpass these baselines in most scenarios, even though some baselines [1, 28] have complete access to the teacher. We also showed ablation results indicating that our evolutionary algorithm is helpful in reducing the distribution gap between the proxy and the true data set. In future work, we would like to turn our attention towards $(i)$ reducing the number of black-box model calls instead of increasing accuracy and $(ii)$ designing preventive solutions, as one of our most important goals is to raise awareness around model stealing, contributing to AI security.

**Acknowledgment**

This work was supported by a grant of the Romanian Ministry of Education and Research, CNCS - UEFISCDI, project number PN-III-P1-1.1-TE-2019-0235, within PNCDI III.

# 6 Broader Impact

Our work has shown that, in the current state of machine learning, it is possible to obtain state-of-the-art results in model functionality replication without knowledge of the internal structure or parameters of the targeted model. With our work, we wish to raise awareness in the domain of AI security, to expose and better understand the exposure and vulnerabilities of public machine learning APIs. Currently, we were able to replicate functionality in a black-box scenario without having knowledge of the training data, reporting a gap of $3.2\%$ between the original model and the replica in one scenario, achieving similar or better results than glass-box approaches. As such, we show that the boundary between glass-box and black-box models is thin and public black-box APIs are as exposed as glass-box models. Our inquiry is that no, or very few models are safe in the current context. We assume that replicas can be analyzed in order to find vulnerabilities in the original model and profit from them. We believe our work represents a solid building block for research in AI security, towards detecting, understanding and preventing any sort of AI vulnerabilities. By exposing techniques such as Black-Box Ripper, we aim to get a head start in designing preventive solutions. Our aim is to stimulate future research in detecting functionality stealing attacks. In the current scenario, for example, the optimization process can be proactively detected based on the API call patterns and stopped. The process can also be impeded by limiting access to prediction confidence scores. Finally, and maybe one of the most important aspects, such model stealing processes can be impeded by designing neural networks that do not focus on textures, but rather on semantics. Our work helps to further push development in this area, which will make deep learning models safer and more robust. Having knowledge of the current work, public APIs can implement such techniques and keep their models in a much safer and desired state, which is the improvement we want to bring to the community. We hope further research on this subject will follow, as we, ourselves, will continue to do so.

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
