[Reviews · NeurIPS 2020]

Review 1

Summary and Contributions: This paper presents Black-Box Ripper, which is a combination of generative-evolutionary algorithms in the teacher-student framework for decoding the latent functionality of the black-box classification models. The investigated black-box problem is considered as zero-shot knowledge distillation. The proposed evolutionary strategy aims at configuring the changes in the output of the generative model such that the generated samples allow to capture some important elements of the teacher. Data generated this way should guide the student to mimic the teacher using the generated data. Several experiments and the ablation study presents the effectiveness of Black-Box Ripper.

Strengths: - The paper presents an intuitive idea of evolving the generated samples for training students that would mimic the teacher. - The method is easy to grasp and results are showing its superiority which is demonstrated by the experiments. - The paper is well written and clear to understand.

Weaknesses: - Assuming a perfect “probability map” (confidence score) at the output of the teacher is a big assumption, since 1) deep classification models suffer from over confidence issues and may be misleading, and 2) coverage of the image space not ensured at all as the fitness does not ensure diversity in the samples obtained. This could lead the evolutionary algorithm to collapse to certain regions of the output space, possibly with misleading probability value, and the results may not be useful to train properly a student mimicking its parent. - The generative models have some problems by itself, like mode collapse (GAN) or blurry effect of VAE, which may result in some deviations in decoding the distribution of the teacher. I am wondering whether the authors have looked at it carefully.

Correctness: Nothing particular to note, the methodology appears fully appropriate.

Clarity: Yes, the paper is well written and the presentation is clear. There is a typo in the abstract (line 9 : “and and”) which can be corrected easily.

Relation to Prior Work: Yes, the similarity and differences of the work is clear.

Reproducibility: Yes

Additional Feedback: ** Post-rebuttal comment: following the reading of other reviews and authors' rebuttal, I feel more confident in the quality of the paper, and as such I upgrade my score to a 7. **


Review 2

Summary and Contributions: In this work, the authors combine Evolutionary Algorithm (EA) and GAN on a teacher-student model for stealing the functionality of the black-box teacher model. Stealing the functionality of black-box model has already been proposed in [1]. Thus, the paper is not novel from the application perspective. In my opinion, the authors simply apply EA on a trained GAN for this application. However, only small datasets are used for evaluation.

Strengths: 1.The combination of GAN and EA seems simple and natural. 2. It seems that the proposed method works better than baselines in the setting with a small number of classes.

Weaknesses: 1. The experimental parts are weak. In the current paper, only a small datasets are employed. It is not sufficient to convince me. I would like to see experiments and comparison with [1] on more practical datasets used in [1]. 2. In the experiments, DeGAN is better than the proposed method on CIFAR100 with 90 classes, and is very competitive on CIFAR100 with 40 and 10 classes. I do have concerns about the proposed method in practical applications with many classes. 3. The influence of the base generative models are not show. Does the GAN used in the paper same as in the baseline [2] [1] Orekondy et al. Knockoff Nets: Stealing Functionality of Black-Box Models. CVPR 2019 [2] Addepalli et al. DeGAN: Data-Enriching GAN for Retrieving Representative Samples from a Trained Classifier. AAAI 2020

Correctness: The experiments are quite weak to support the claim that the proposed method are better than other baselines.

Clarity: The paper is well written.

Relation to Prior Work: More discussion about the relation to other methods for stealing functionality of black-box models ( [1] [3] ) is better. [3] Krishna et al. Thieves on Sesame Street! Model Extraction of BERT-based APIs

Reproducibility: Yes

Additional Feedback: The paper can be imporved by including experiments and comparison with baseline on more practical dataset. ======================================================================================================================= POST REBUTTAL: Thanks to the authors' feedback. My main concerns on the dataset and results in many classes have been addressed. However, I still concern about the novelty of the paper. Black-box model stealing is also discussed in [1]. Considering previous work[1] in the context, using EA on a pre-trained GAN for model stealing is not so novel. The novelty is limited in using EA in this task. It is better for the authors to discuss the contribution compared with [1]. Nevertheless, I agree that it may have a certain contribution to the improvement in this application. So I decided to increase my score.


Review 3

Summary and Contributions: The paper presents an original approach to model-stealing, or knowledge distillation in which the stolen model is a complete black-box (but the output probabilities are fully known) and the training data is not available. The trik is to learn a generative model offline, and to use an evolutionary algorithm to generate data that are classified with high confidence by the black-box into given classes, and to train the student model on these data. Reported results perform well, and often better than competitor approaches.

Strengths: An original idea tackling a difficult problem (black-box teacher and unknown training data) that seems to performs very well, demonstrating that a very simple evolutionary approach can make its way through low-dimensional latent spaces. Quite significant imo, and highly relevant to NeurIPS.

Weaknesses: The main weakness is the lack of statistical study (showing statistics over several independent runs, and not the results of a single run - see below). Also, you use an arbitrary fixed standard deviation for the Gaussian noise you add in the mutation. Wouldn't it help to perform adaptive mutations, like routinely done in the Evolution Strategies field? Finally, you only evolve for 10 generations (which might explain BTW that adaptive mutations would not really have time to set up)" Did you try evolving longer, i.e., getting better performing images as far as raw classification is concerned?

Correctness: Empirical methodology is correct, except that there are no statistical study of the results: whereas Addepali et al. repeated their experiments 5 times, the work here seems to present the results of a single run. I don't understand why. Furthermore, this is not acceptable for stochastic algorithms like evolutionary ones. Also, even different generative models could be used, trained on different samples of the proxy datatset.

Clarity: Yes, the paper is very clear, and easy to follow. General figure and algorithm in pseudo-code help a lot.

Relation to Prior Work: Nice review of existing approaches - though I am not expert in these domains (model stealing and knowledge distillation) so I wouldn't probably be aware of recent work in these areas that the paper could have missed.

Reproducibility: No

Additional Feedback: I am satisfied with the answers to my comments in the rebuttal, and that reinforces my "accept" recommendation.


Review 4

Summary and Contributions: An evolutionary algorithm is used to optimize the generator so that the generator can generate a well-classified sample of the teacher network. The proxy dataset is used to approximate the sample of the dataset used for teacher network training through the generator. In this paper, the teacher network is treated as a complete black box model, and the student network achieves some performance.

Strengths: There is a certain theoretical basis. The paper has a certain novelty, but not so novel, because evolutionary algorithms have been widely used. The performance improvement of the model is not obvious compared with other methods. But given that the model treats the stolen model as a completely black box model, it is acceptable for the model to achieve current performance.

Weaknesses: There are few comparative models in this paper. And this paper is not innovative enough. The detailed description of the evolutionary algorithm used in this paper is not clear, such as what kind of coding method is used. And there are multiple methods for evolutionary algorithms, why do you choose this kind of evolutionary algorithm.

Correctness: The claims and method are correct and the empirical methodology is correct.

Clarity: No, the description of the evolutionary algorithm in this paper is not clear.

Relation to Prior Work: yes

Reproducibility: No

Additional Feedback: Describe the evolutionary algorithm in detail and add comparative experiments.

[Author Response · NeurIPS 2020]

**Rebuttal for 3893: Black-Box Ripper: Copying black-box models using generative evolutionary algorithms**

We thank reviewers for their useful comments and insights. The reviewers appreciated our idea as intuitive and original
(R1, R3), our method as easy to grasp (R1), significant and highly relevant to NeurIPS (R3), our results as showing the
superiority of our method (R1, R3) and our paper as well-written and clear (R1, R2, R3). Furthermore, R3 appreciated
our nice review of existing approaches. We next address concerns raised by the reviewers.

**R1:** The evolutionary algorithm might collapse to certain regions of the output space. **Answer:** We have analyzed
the pixel-level variability of 1000 random images generated by GAN and 1000 random images optimized by our
evolutionary strategy, the resulting means of the normalized pixel-level standard deviations being 0.26 and 0.27,
respectively. This demonstrates that the proposed evolutionary strategy does not collapse to certain regions.

**R1:** Did the authors look at mode collapse of GAN or blurry effect of VAE? **Answer:** The GANs used in our
experiments did not suffer from mode collapse. However, for the second set of experiments, the considered VAE
does indeed output blurry images. Nonetheless, aspects such as the quality of the images generated by GAN or VAE
are not relevant to our approach, as long as we are able to achieve state-of-the-art performance levels in stealing the
functionality of black-box models, as shown in Tables 1, 2, 3 from our paper.

**R2:** The experimental parts are weak, only small data sets being employed. **Answer:** First, we considered the same
data sets as Addepalli et al. [1], our main competitor, their paper presenting results on such data sets being published at
AAAI 2020. Aside from the experimental setup used by Addepalli et al. [1], in which the images have low resolution,
we have performed additional experiments using larger images from CelebA-HQ, ImageNet Cats and Dogs and 10
Monkey Species. In the experiments presented in Table 3 (see paper), the teacher and student models take images of
$224 \times 224$ pixels as input. We thus believe that our experiments cover a wider range than those of Addepalli et al. [1].

**R2:** Concerns about the proposed method in practical applications with many classes. **Answer:** The experiments
performed on higher-resolution image shows that our method works well when confronted with a large multi-class
latent space, considering that ImageNet Cats and Dogs contains 143 classes. Regarding our results on 90 and 40 classes
from CIFAR-100, we note that our accuracy rates are comparable to those reported by Addepalli et al. [1]. However,
our method is significantly more general, as their method only works in white-box scenarios, requiring complete access
to the network to perform back-prop. Our method works in a black-box setup, requiring no knowledge of the internal
structure or parameters of the model.

**R2:** Is the GAN used in the paper same as in the baseline [1]? **Answer:** The GANs have equivalent architectures, but
the models are not identical, mainly because Addepalli et al. [1] train the GAN along with the student, whereas we only
use a pre-trained GAN.

**R3:** The main weakness is the lack statistics over several independent runs. **Answer:** We note that Addepalli et al. [1]
typically reported results of independent runs, with one exception. They performed 5 runs for a single scenario, in
which 10 classes from CIFAR-100 are used as proxy. We hereby present results for 5 runs of our approach for the same
case in Table 1, demonstrating the stability and superiority of our approach over that of Addepalli et al. [1]. Upon
acceptance, we will include mean and standard deviations over 5 runs for all our experiments in the final paper.

Table 1: Accuracy rates (in %) for 5 runs on CIFAR-10 as true data set and 10 classes of CIFAR-100 as proxy.

| Method | Run 1 | Run 2 | Run 3 | Run 4 | Run 5 | Mean $\pm$ Std. Dev. |
|---|---|---|---|---|---|---|
| DeGAN [1] | 66.9 | 74.6 | 72.6 | 76.6 | 71.6 | $72.6 \pm 3.26$ |
| Black-Box Ripper (Ours) | 78.2 | 78.0 | 77.4 | 77.9 | 78.2 | $77.9 \pm 0.29$ |

**R3:** Wouldn't it help to perform adaptive mutations? **Answer:** In a set of preliminary experiments, we tried using
CMA-ES, without observing any performance improvements. Hence, we decided to stick to the most straight forward
method that already outperforms existing approaches.

**R3:** Authors only evolve for 10 generations. **Answer:** In a set of preliminary experiments, we tested with up to 50
generations, but we observed that the evolved exemplars typically converge in 10 generations or even less. To reduce
the computational time, we decided to keep only 10 generations, which seems sufficient (see Tables 1, 2, 3 in paper).

**R5:** The detailed description of the evolutionary algorithm used in this paper is not clear, such as what kind of coding
method is used. **Answer:** The coding method is straight forward: our data samples are input noise vectors from the
latent space of the generator. Upon acceptance, we will strive to further improve the clarity of our presentation.

**R5:** Why do you choose this kind of evolutionary algorithm? **Answer:** We tried the simplest evolutionary algorithm,
which already seems capable of surpassing glass-box approaches. Further tweaking the evolutionary algorithm can only
bring improvements, putting an even greater gap between our results and those of the state-of-the-art methods.

**R5:** Paper is not innovative enough. **Answer:** Model stealing has never been studied or proven possible with
evolutionary algorithms. We show that our evolutionary strategy surpasses other, more relaxed, glass-box state-of-the-
art methods. Furthermore, the originality of our idea is appreciated by R1, R3.

[1] Addepalli et al., DeGAN: Data-Enriching GAN for Retrieving Representative Samples from a Trained Classifier.
AAAI 2020.

[Meta-Review · NeurIPS 2020]

The basic idea of evolving a data set that mimics a black box model of a teacher is intuitive and interesting, although not entirely novel. But the combination of EA and GAN algorithms for realizing a solution to this problem is. The experimental results presented show superiority over the tested baselines, and the paper is well written and easy to understand. One limitation is that the method only evaluated on small data sets and the description of the EA that is used needs to be better explained. The options of several reviewers were raised as a result of clarifications provided in the user response, and the consensus recommendation on this paper is to accept. Please be sure to attend to the reviewer comments as you prepare your final version.